# The first ice-free day in the Arctic Ocean could occur before 2030

Céline Heuzé ●[1,3] ✉ & Alexandra Jahn ●[2,3] ✉

Projections of a sea ice-free Arctic have so far focused on monthly-mean ice-free conditions. We here provide the first projections of when we could see the first ice-free day in the Arctic Ocean, using daily output from multiple CMIP6 models. We find that there is a large range of the projected first ice-free day, from 3 years compared to a 2023-equivalent model state to no ice-free day before the end of the simulations in 2100, depending on the model and forcing scenario used. Using a storyline approach, we then focus on the nine simulations where the first ice-free day occurs within 3–6 years, i.e. potentially before 2030, to understand what could cause such an unlikely but high-impact transition to the first ice-free day. We find that these early ice-free days all occur during a rapid ice loss event and are associated with strong winter and spring warming.

The observed decline of the Arctic sea ice cover[1] is expected to continue in the future[2]. The potential of an ice-free Arctic Ocean is one of the most striking examples of the ongoing anthropogenic climate change, with a visible transition from a white Arctic Ocean to a predominantly blue Arctic Ocean during the summer[3]. While the first occurrence of ice-free conditions has primarily symbolic significance, a transition to an Arctic Ocean that regularly has a sea ice area of less than 1 million km² (commonly used as the ice-free threshold[4–7]) in the summer is expected to have cascading effects on the rest of the climate system: It would notably enhance the warming of the upper ocean, accelerating sea ice loss year round[8] and therefore further accelerating climate change[9], and could also induce more extreme events at mid-latitudes[10]. A further reduction of the summer sea ice cover will also negatively impact the already-stressed Arctic ecosystem, from the emblematic polar bear to the crucial zooplankton[11,12].

Current projections from climate models suggest that the first monthly mean September sea ice area (SIA) at or below 1 million km² could occur by 2050[2], but predictions of an ice-free Arctic have large uncertainties, due to model biases and internal variability[7,13,14]. However, before the September monthly mean reaches the ice-free threshold, we will see a first ice-free day with a SIA of 1 million km² or less[7]. Given that multi-model projections of the first ice-free day are so far lacking[7], when such first ice-free days could occur is currently unknown. To fill this gap, we here present the first projection of daily ice-free conditions, based on daily CMIP6[15,16] sea ice concentration data.

We here focus on predictions of ice-free conditions relative to a 2023-equivalent model state with a daily SIA minimum equal or larger than the 3.39 million km² that were observed in 2023[17], to illustrate how long it could take to transition from a daily SIA minimum similar to the one observed in 2023 to the first day with a daily SIA of 1 million km² or less (see the Section "Methods" for details on the exact definition of the 2023 equivalent conditions). Furthermore, this also removes some of the uncertainty in model simulations, by re-calibrating models to a common starting SIA state. In Section "Timing of the first ice-free day" we use 11 different CMIP6 models that performed best over the historical period (see Section "Methods") with a total of 366 ensemble members to provide probabilistic predictions of first daily ice-free conditions under different CMIP6 future emission scenarios (Shared Socio-economic Pathways - SSPs), and show how much earlier they occur than monthly first ice-free predictions.

In Sections "Storylines: From 2023 equivalent conditions to the first ice-free day in 3–6 years" to "Final-year triggers: Winter warm air intrusions, spring blocking, and summer storms", we then focus on understanding the evolution of the simulations that reach ice-free conditions the fastest from their 2023 equivalent states, following a storyline approach[18]. By focusing on these quick transition simulations, we are not suggesting that ice-free conditions will be reached this

[1]Department of Earth Sciences, University of Gothenburg, Box 460, 405 30 Göteborg, Sweden. [2]Department of Atmospheric and Oceanic Sciences and Institute of Arctic and Alpine Research, University of Colorado Boulder, Boulder, CO, USA. [3]These authors contributed equally: Céline Heuzé, Alexandra Jahn. ✉e-mail: celine.heuze@gu.se; alexandra.jahn@colorado.edu

quickly. Instead, the goal is to raise awareness for the potential of a rapid loss of sea ice in the near-future, and to provide insights into what may lead to such rare but high-impact events.

## Results

### Timing of the first ice-free day

The earliest ice-free day occurs 3 years after 2023-equivalent conditions, based on the 11 analyzed CMIP6 models (under SSP1-2.6, Fig. 1a and Table 1). Another two CMIP6 models show the earliest ice-free day within 4 years (also under SSP1-2.6, 1a), and another 6 ensemble members go ice-free within 5-6 years (under SSP1-2.6 to SSP3-7.0, Table 1). Overall there are 34 ensemble members from four different models reaching the first ice-free day within 10 years (under SSP1-2.6 through SSP5-8.5). Notably, the emission scenario does not play an important role here, with the ensemble members that reach ice-free conditions within 10 years from 2023-equivalent conditions occurring under all scenarios except the lowest emission scenario (SSP1-1.9, Fig. 1a). In fact, the fastest

three transitions (in 3–4 years) occur under SSP1-2.6, the second lowest CMIP6 forcing scenario. This clearly shows that these rapid transitions from 2023-equivalent conditions to the first ice-free day occur primarily due to internal variability, not due to the strength of the external forcing. The large influence of internal variability on the earliest ice-free days agrees with findings for the earliest ice-free month[2,7,14].

That said, we do find that under the weaker forcing scenarios (SSP1-1.9 and SSP1-2.6), there is a possibility of seeing a delay in the first ice-free day that is not seen in the majority of the higher emission simulations (Fig. 1a, b). Specifically, we find that the first ice-free day could occur over 50 years after the 2023 equivalent under the analyzed SSP1-1.9 and SSP1-2.6 simulations, or may not reach ice-free conditions at all before the end of the 21st century scenario simulations (Fig. 1b). Furthermore, under SSP1-1.9 the earliest ice-free day occurs 18 years after 2023 equivalent conditions (Supplementary Table S1), while under all other scenarios at least one ensemble member reaches ice-free conditions within less than 10 years (Fig. 1a).

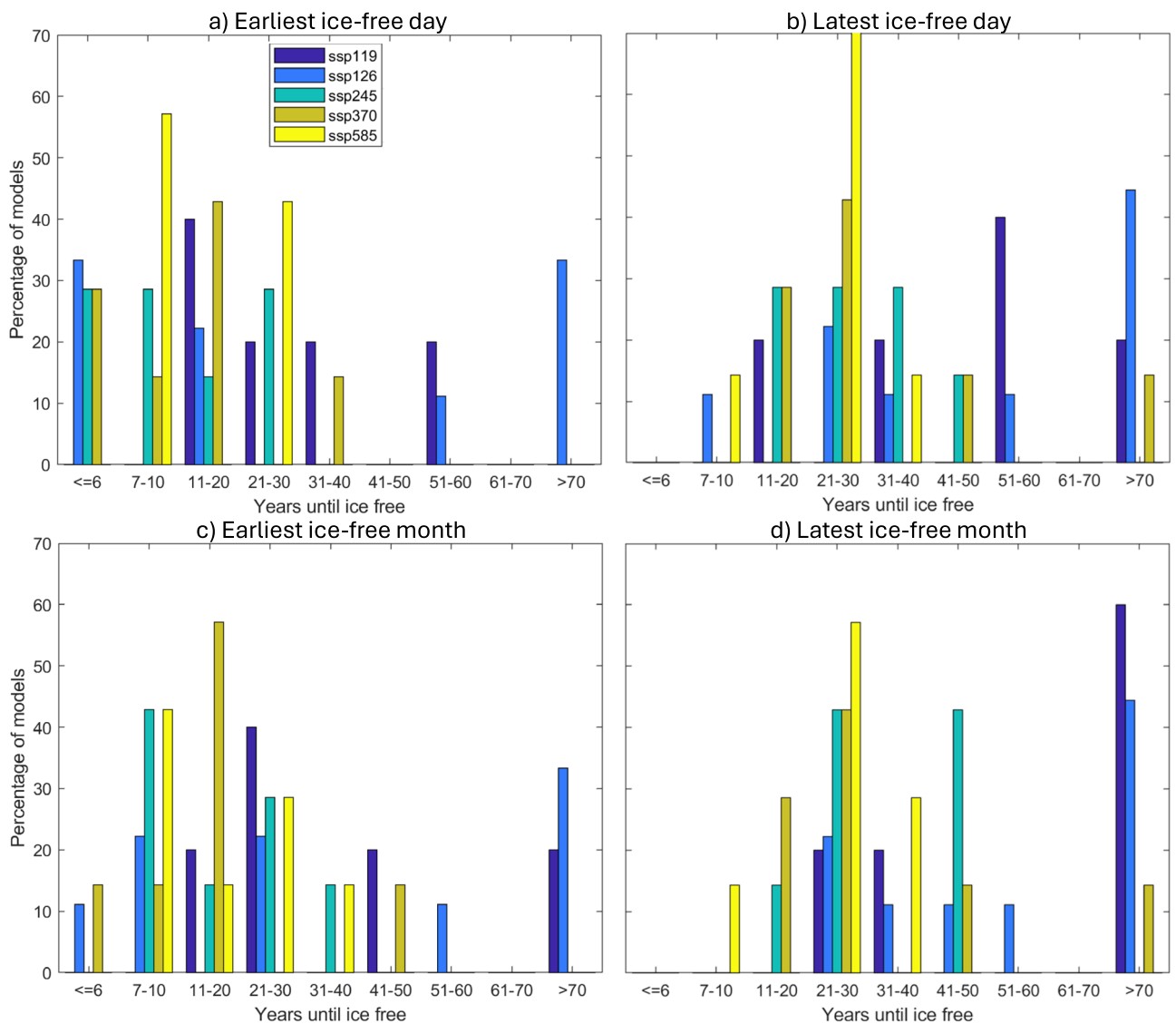

**Fig. 1 | Time to ice-free conditions.** Histograms of time from 2023 equivalent daily sea ice area (SIA) minimum until the first ice-free day (**a**, **b**) or ice-free month (**c**, **d**), for the earliest (**a**, **c**) and latest (**b**, **d**) members of each model for all forcing scenarios (Shared Socioeconomic Pathways, SSP). For models that only had one ensemble member available, the same ensemble member is shown in the earliest and latest histograms. This figure shows that the earliest ice-free day does not depend on the forcing scenario but rather on internal variability, that the strength of the forcing only impacts whether we see models without any ice-free conditions, and that the overall distribution of the earliest and latest first ice-free days and months are generally similar but with generally a slightly delay of the the first ice-free month versus day. See Supplementary Table S1 for details on which models reach an ice-free day when and which models have only one ensemble member. The same is shown for the first ice-free month in Supplementary Table S2.

Once a first ice-free day occurs in the CMIP6 simulations, it lasts between one day and 71 days, with a mean length of 25 days and a standard deviation of 17 days. As the duration of the first ice-free occurrence in the daily data varies so strongly, the number of years between the first ice-free day and the first ice-free month also varies strongly among the simulations (Supplementary Fig. S1), ranging from occurring in the same year to several decades later, or not reaching monthly-mean ice-free conditions at all. For the earliest ensemble members, there is some scenario dependence in how fast the first ice-free month follows a first ice-free day (Supplementary Fig. S1a), but for the latest ensemble member the first ice-free month follows quickly after the first ice-free day under all SSPs except the SSP1-1.9, which shows a delay of up to several decades for some models. The distribution of the earliest and latest ice-free month is generally similar to the distribution of the earliest and latest ice-free day (Fig. 1c and d versus a and b), with generally fewer early ice-free months than ice-free days and more simulations under the two lowest SSPs not reaching monthly ice-free conditions at all. We find that the ensemble member with the earliest ice-free day is also the one with the earliest ice-free month in just over half of the assessed SSP scenarios with more than one ensemble member, but there are also many SSP scenarios where a different ensemble member reaches monthly ice-free conditions before the member that had the first ice-free day gets there (compare Supplementary Table S2 to Supplementary Table S1).

Overall, this analysis shows that a reduction in anthropogenic warming to the level of the SSP1s (which means staying around or under 1.5 °C of global warming[19]) may not prevent an internal-variability induced first ice-free day but could increase the probability of delaying or avoiding an ice-free day and month.

## Storylines: from 2023 equivalent conditions to the first ice-free day in 3–6 years

To illustrate how we could potentially transition from a seasonal daily SIA minimum comparable to what was observed[17] in 2023 (3.39 million km$^2$) to the first daily ice-free conditions within just a few years, we will now focus on the simulations that do so in 3–6 years - referred to in the following as quick transition simulations. This includes nine simulations, one simulation that reaches ice-free conditions within 3 years, two that take 4 years, two that take 5 years, and four that take 6 years, spanning four different climate models (ACCESS-CM2, CanESM5, EC-Earth3, and MPI-ESM1-2-LR) and three emission scenarios (SSP1-2.6, SSP2-4.5, SSP3-7.0, see Table 1). Note that for all of these models that have quick transition members, there are other members from the same model that take much longer to reach first daily ice-free conditions under the same forcing (with a maximum offset of 58 years, Supplementary Table S1).

The reason for that large difference between ensemble members from the same model is that the effect of internal variability on Arctic sea ice is large[20–22]. Thus, it is not that these four models (ACCESS-CM2, CanESM5, EC-Earth3, and MPI-ESM1-2-LR) are generally fast to lose their SIA; it is the specific evolution of the internal climate variability in these specific ensemble members that leads to the quick transitions from a 2023-equivalent SIA daily minimum to daily ice-free conditions within 3–6 years. Specifically, only 1% of the MPI-ESM1-2-LR simulations (2 out of 205) are quick transition simulations, 3% of the CanESM5 simulations (2 out of 72), 5% of the ACCESS-CM2 simulations (2 out of 39), and 30% of the EC-Earth3 simulations (3 out of 10; see Supplementary Table S5).

## Multiyear sea ice area transition to the first ice-free day

By definition, all the quick transition simulations have a very rapid transition from a 2023 equivalent daily minimum SIA of 3.39 million km$^2$ to daily ice-free conditions (Fig. 2). To put this rapid transition into context of observed sea ice changes, this kind of decrease in the daily minimum is equivalent to less than three times the observed difference

**Table 1 | Characteristics of the nine quick transition simulations: Showing the time from 2023 equivalent conditions to the first ice-free day, the degree of global warming for the year of the first ice-free day compared to pre-industrial period (5-year running mean, see Section "Methods"), the date of that first ice-free day, and the duration of that first ice-free period (in days)**

| Model & member & SSP | Time to first ice-free day | Warming by that year | First ice-free day | Ice-free period |
|---|---|---|---|---|
| ACCESS-CM2 r6i1p1f1 SSP3-7.0 | 6 years | 1.7 °C | Aug-11 | 53 days |
| ACCESS-CM2 r7i1p1f1 SSP1-2.6 | 4 years | 1.5 °C | Sep-09 | 12 days |
| CanESM5 r8i1p1f1 SSP1-2.6 | 4 years | 2.3 °C | Aug-20 | 25 days |
| CanESM5 r9i1p1f1 SSP1-2.6 | 6 years | 2.5 °C | Aug-15 | 42 days |
| EC-Earth3 r4i1p1f1 SSP1-2.6 | 3 years | 1.7 °C | Aug-26 | 25 days |
| EC-Earth3 r8i1p1f1 SSP1-2.6 | 5 years | 1.6 °C | Aug-29 | 17 days |
| EC-Earth3 r12i1p1f1 SSP2-4.5 | 5 years | 1.5 °C | Aug-14 | 32 days |
| MPI-ESM1-2-LR r38i1p1f1 SSP3-7.0 | 6 years | 1.7 °C | Sep-03 | 11 days |
| MPI-ESM1-2-LR r43i1p1f1 SSP2-4.5 | 6 years | 1.5 °C | Aug-27 | 23 days |

*SSP* stands for Shared Socioeconomic Pathway.

between the 2011 and 2012 daily SIA minimums, which is not even the largest year-to-year difference between daily minimums observed. Notably, the rapid sea ice loss in all quick transition simulations meet the criteria for a Rapid Ice Loss Event (RILE)[23–26] during September, with all of them reaching daily ice-free conditions during a RILE period, defined as an at least 4-year period where the average trend in the 5-year running mean sea ice extent (SIE) is larger than − 0.3 million km$^2$ per year[25] (see the shading in Fig. 2). In addition, all of them show a RILE or near RILE (if relaxing the RILE threshold from − 0.3 to − 0.299 million km$^2$ per year) in August during the transition period from the 2023 equivalent conditions to the first ice-free day (Supplementary Fig. S2). While the large reduction in the minimum SIA towards the first ice-free day starts in the 2023 equivalent year in all quick transition simulations (Fig. 2), in some quick transition simulations the RILE events in September and August (Supplementary Fig. S2) already start slightly before the 2023 equivalent year, due to the definition of RILEs based on the trends in the 5-year running mean SIE. For September, the RILEs start 1.7 years before the 2023 equivalent in the mean, with a median of 1 year, while the August RILEs or near-RILEs start 1 year prior in the mean and median, with a range of 3 years before and 2 years after. That means that while all of the quick transition simulations reach the first ice-free conditions during a RILE, the length of the RILE and the start of the RILE relative to the 2023 equivalent conditions varies among the 9 simulations.

As their name implies, RILEs describe a rapid loss of sea ice over a short period of time[23]. RILEs have been found in nearly all CMIP6 models and all months of the year[26]. The minimum required sea ice extent loss during a RILE exceeds even the September SIA loss observed in the early 21st century[26], even though the 4-year period between 2005 and 2007 got close, with the 4-year trend in the 5-year running mean sea ice extent in September reaching − 0.23 million km$^2$ in observations (compared to the −0.3 million km$^2$ threshold for a RILE). A different sequence of observed September SIE minima between 2006 and 2012, however, could have led to a RILE. In particular, a RILE would have been observed if we had seen the two record September SIE drops leading to the 2007 and 2012 minima within a 4

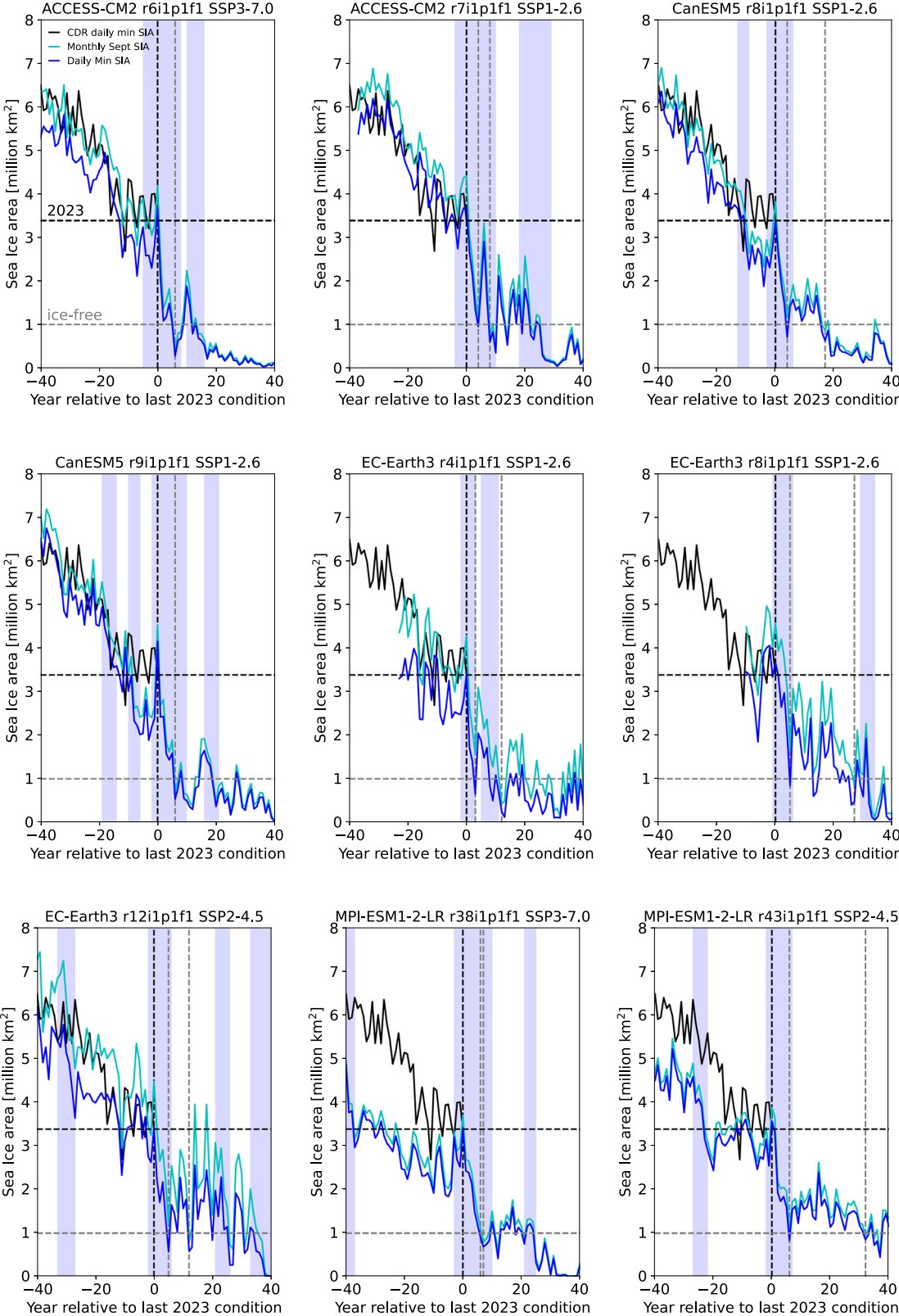

**Fig. 2 | Rapid transition from 2023 equivalent state to the first ice-free day.** The simulated sea ice area (SIA) daily minimum (blue), simulated September monthly mean simulated SIA (cyan), and observed daily minimum SIA based on the CDR SIC data[17] (black) [all in million km²]. Also shown are Rapid Ice Loss Events (RILEs) during September (shaded), defined based on the monthly sea ice extent[25]. A vertical black dashed line indicates the year when the simulated daily SIA minimum was last at or above the 2023 observed daily SIA minimum of 3.39 million km², which is in turn is indicated with a horizontal black dashed line. Vertical dashed gray lines show when ice-free conditions are reached (first for daily, then for monthly, if there is only one both reach ice-free conditions for the first time in the same year). The gray dashed horizontal line shows the 1 million km² ice-free threshold. Supplementary Fig. S2 shows that RILEs or near-RILES also occur during August. This illustrates that the transition to first daily ice-free conditions occurs during a RILE in the quick transition simulations.

year period instead of within the 6 year period they occurred in. This shows that RILEs events are possible with observed year-to-year sea ice changes. CMIP6 models suggest that RILEs are most likely to start when the Arctic September mean SIE is at slightly less than 4 million km²[26]; since the 2023 September mean SIE was 4.24 million km², it is not unexpected that we have not yet seen a RILE.

While the exact drivers of RILEs are still under investigation, both atmospheric and oceanic drivers have been suggested as being important[25]. The atmosphere was found to be especially important in driving RILEs once the sea ice cover is already primarily limited to the deep Arctic Ocean basin[25], as is the case during the summer on the way to the first ice-free day.

The rapid transition to a first early ice-free day, however, does not just occur in the summer, but also includes a reduction of the sea ice cover during the autumn, winter, and spring (Fig. 3a, b). While this is clear from the SIA alone in most of the quick transition models (Fig. 3a), the EC-Earth3 simulations do not show a notable change in the wintertime SIA. But when taking into account the sea ice thickness as well, and thus looking at the total sea ice mass (Fig. 3b), it is clear that in all models we see a clear decrease in the total Arctic sea ice mass year-around. This means that reduced sea ice thickness in the wintertime occurs in all models soon (within 1 or 2 years) after they had a SIA minimum at or above the 2023 conditions, even when it is not apparent in the wintertime SIA. As discussed in the next section, this reduction in sea ice is linked to warm winters and springs (Supplementary Fig. S3), as well as a delayed freeze up in the previous autumn[27] (Fig. 3b).

Once the first ice-free day is reached in these quick transition simulations, the Arctic does not remain ice-free for one day only. The ice-free period lasts between 11 and 53 days in the 9 quick transition simulations, with an average duration of 27 days (Table 1). The ice-free duration is set primarily by the day the first ice-free conditions occur, with the simulations that show ice-free conditions earliest showing the longest duration. Specifically, we find that the first ice-free day for the quick transition simulations ranges between Aug 11 to Sept 09 (days 223–252), with an average of Aug 26 (day 236). The first ice-free day occurs in September for ACCESS r7i1p1f1 and MPI-ESM1-2-LR r38i1p1f1, which also have the shortest ice-free duration (12 and 11 days, respectively).

Notably, in the year with the first ice-free day in the quick transition simulations, the 2023 daily SIA minimum value of 3.39 million km² is reached at the latest by July 31st (day 212) (see Fig. 3a) - 42 days earlier than the observed 2023 daily minimum on Sept 11 (day 254, according to the Climate Data Record (CDR) derived SIA[17]). Thus, if in the future the observed SIA crosses the 3.39 million km² SIA already in July, this could be a warning sign that an ice-free day may occur later in the summer.

## Final-year triggers: winter warm air intrusions, spring blocking, and summer storms

For all quick transition simulations, the sea ice is pre-conditioned for an ice-free day: Most years leading up to the year of the first ice-free day have a delayed atmospheric cooling in autumn and warm spells all the way to December (Supplementary Fig. S3 and Supplementary Table S3), consistent with the delayed and reduced sea ice formation described above. A series of events in the last winter, spring, and summer finish weakening the ice both dynamically and thermo-dynamically, leading to that first ice-free day.

For all cases, the last winter is warm (Fig. 4a and Supplementary Fig. S3). This final winter warmth is anomalous compared to the previous years over the Central Arctic Ocean for all cases (Supplementary Table S3), except for EC-Earth3 r4i1p1f1 which has a strong warm anomaly over the Barents Sea instead (heating degree day difference of 148 HDD). North of 80°N, maximum air temperatures exceed the spring transition temperature of −20 °C[28] all winter long, most often in association with strong high pressures (Fig. 4b, c, and Supplementary Fig. S4), but also sometimes in association with strong low pressures, i.e. due to a warm air intrusion (Supplementary Fig. S5a). The warmth and high pressure persist into the spring for all nine simulations, with two different patterns: A year where the spring warming is shifted up to one month early (see e.g. the case that becomes ice-free fastest, EC-Earth3 r4i1p1f1, Supplementary Fig. S3), or a year that is not extreme but is more stable, has fewer cold spells than usual (see e.g. CanESM5 r9i1p1f1, Supplementary Fig. S3 and Supplementary Table S3). Heat-waves with maximum temperatures exceeding 0 °C are common in that last spring (Fig. 4), lasting for several days because the warm air is blocked over the central Arctic by a high pressure system (Supplementary Fig. S5b).

The last summer is warm to very warm for all quick transition simulations, with temperatures that can exceed 10 °C from day 151 (late May, Fig. 4). The atmospheric pressure becomes less stable, and in six out of nine simulations (CanESM5 r8i1p1f1, all three EC-Earth3 and both MPI-ESM1-2-LR), storms cross through the Arctic, especially so in the

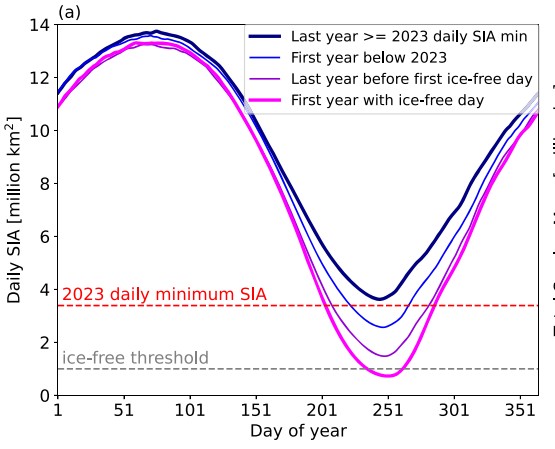
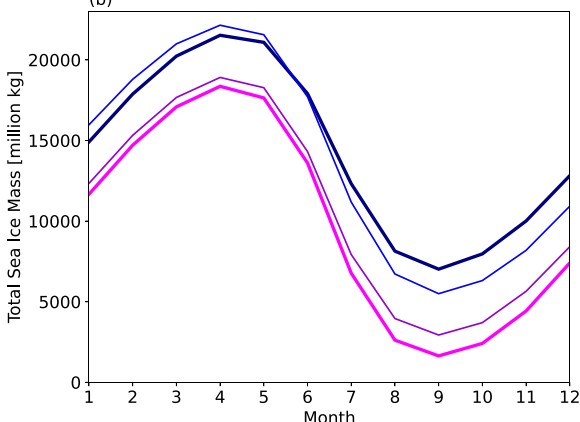

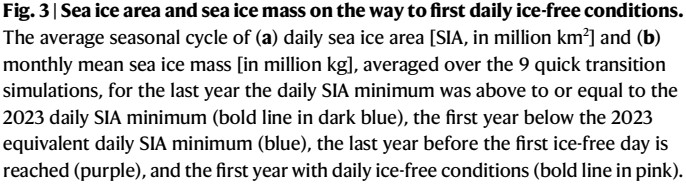

**Fig. 3 | Sea ice area and sea ice mass on the way to first daily ice-free conditions.** The average seasonal cycle of (**a**) daily sea ice area [SIA, in million km²] and (**b**) monthly mean sea ice mass [in million kg], averaged over the 9 quick transition simulations, for the last year the daily SIA minimum was above to or equal to the 2023 daily SIA minimum (bold line in dark blue), the first year below the 2023 equivalent daily SIA minimum (blue), the last year before the first ice-free day is reached (purple), and the first year with daily ice-free conditions (bold line in pink). The 2023 daily SIA minimum (3.39 million km² based on ref. 17) is shown as red dashed horizontal line and the 1 million km² ice-free line is shown as gray dashed horizontal line. This figure shows that the sea ice loss leading up to the first ice-free day is not limited to only the daily minimum SIA getting lower, but occurs throughout the entire year and is especially clear in the sea ice mass (**b**), indicating a large reduction in wintertime sea ice thickness in addition to a reduction in sea ice area.

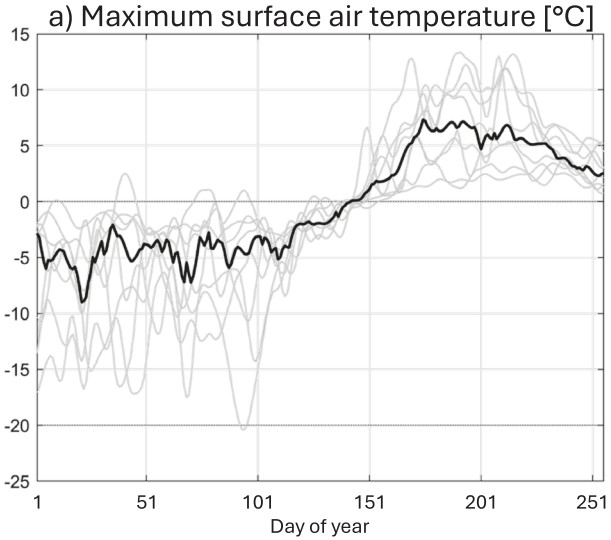

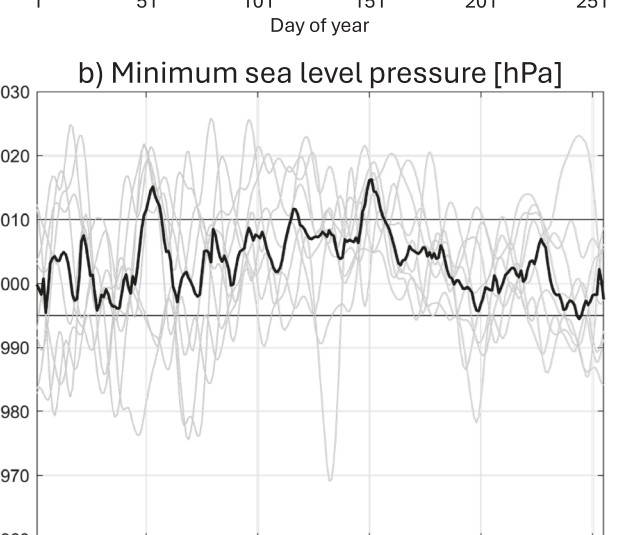

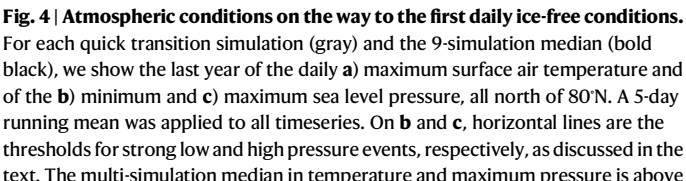

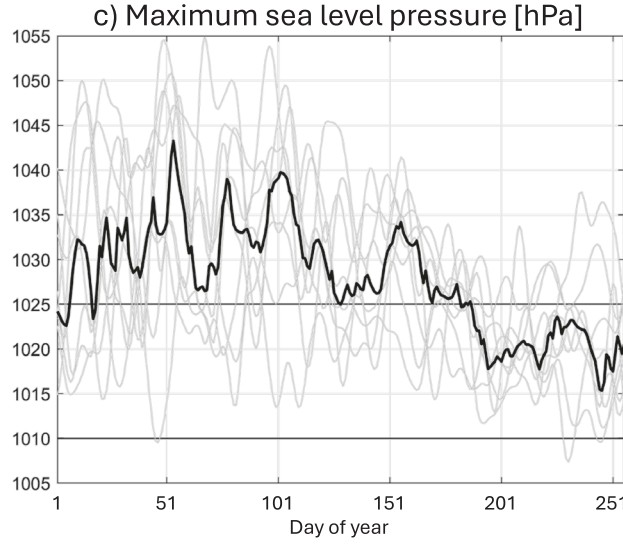

**Fig. 4 | Atmospheric conditions on the way to the first daily ice-free conditions.**
For each quick transition simulation (gray) and the 9-simulation median (bold black), we show the last year of the daily **a**) maximum surface air temperature and of the **b**) minimum and **c**) maximum sea level pressure, all north of 80°N. A 5-day running mean was applied to all timeseries. On **b** and **c**, horizontal lines are the thresholds for strong low and high pressure events, respectively, as discussed in the text. The multi-simulation median in temperature and maximum pressure is above these thresholds, highlighting the persistence of the heatwaves and blocking patterns. The multi-simulation median in minimum pressure in contrast is not particularly informative, as the simulations have short-lived storms at different times. This figure illustrates that all quick transition simulations are warm in winter, in association with extreme low (warm air intrusions) or high (blocking) pressure events. They are very warm in spring, and many become stormy in summer.

last month or even last days before the first ice-free day (Fig. 4). EC-Earth3 r4i1p1f1, which has an ice-free day only 3 years after 2023 conditions, has one extensive storm shooting from the Kara Sea region to the Canada basin in 5 days, culminating in the earliest simulated first ice-free day (Supplementary Fig. S6). In most of the quick transition simulations, such as EC-Earth3 r12i1p1f1 (Supplementary Fig. S5c), several weak storm systems simultaneously stress the sea ice at various locations across the Arctic in the days leading up to the first ice-free day. For six out of 9 simulations, this summer warmth over the Central Arctic Ocean is anomalous when compared to the other years since 2023-equivalent (Supplementary Table S3). The other three have no significant difference over the Central Arctic but are instead anomalous over the surrounding seas:

- ACCESS-CM2 r7i1p1f1 is anomalously warm over the Barents Seas (anomaly of 71 HDD, Supplementary Table S4);
- CanESM5 r9i1p1f1 is anomalously warm over the Laptev, East Siberian and Chukchi seas (anomaly of 87, 28 and 41 HDD, respectively, Supplementary Table S4), while being anomalously

stormy over the Central Arctic Ocean (anomalies in both extremes of SLP are negative, Supplementary Table S3);
- MPI-ESM1-2-LR r38i1p1f1 is anomalously warm over the Kara Sea (anomaly of 80 HDD, Supplementary Table S4), and likewise, is anomalously stormy over the Central Arctic Ocean (Supplementary Table S3).

The warm atmospheric conditions triggering the first ice-free day are predicted to become increasingly common in a warmer world[29]. As the Arctic warms, heatwaves at any season become more likely[30], as do warm air intrusions and storms[31]. But it is not too late to avoid an ice-free day: For all quick transition cases, the first ice-free day occurs in years at or above the 1.5 °C of global warming above pre-industrial, which is the target to not exceed set by the Paris Agreement[32] (Table 1). This agrees with prior work on the first monthly ice-free Arctic, which also found that ice-free conditions may be avoided if global warming stays below the Paris Agreement target of 1.5 °C[33–35].

## Discussion

The first time the Arctic reaches ice-free conditions will be an event with a high symbolic significance, as it will visually demonstrates the ability of humans to change one of the defining features of the Arctic Ocean through anthropogenic greenhouse gas emissions - the transition from a white Arctic Ocean to a blue Arctic Ocean[3]. So far, all multi-model predictions have focused on predicting the first ice-free conditions in the monthly mean values[2,4,5,14,33,34]. But the first time we will observe ice-free conditions in the Arctic Ocean will be in the daily satellite data, not in a monthly mean product[7]. Thus, to set realistic expectations as to when we could first observe ice-free conditions in the Arctic, we here used daily data from CMIP6 models to provide the first multi-model predictions of the first ice-free day.

We showed that the earliest ice-free day in the Arctic Ocean could occur within 3 years from 2023 sea ice area (SIA) minimum equivalent conditions, i.e. that there is a non-zero probability of an ice-free day before 2030. The highest probability of the earliest ice-free day occurring lies within 7–20 years, based on the earliest ensemble member from all SSPs from the 11 CMIP6 models analyzed (Fig. 1). Across all 366 simulations from all SSPs, the median first ice-free day occurs within 24 years with a mean at 29 years. Note that all of these projections start from the last time the daily SIA minimum is above or equal to 3.39 million km². That could be in 2023, if all daily SIA minima after 2023 are below 3.39 million km². But the countdown to the first ice-free day could also start from a future year, if the observed daily SIA minimum in years after 2023 is above 3.39 million km².

Similar to the first monthly ice-free predictions[7], there is a large prediction uncertainty of the first ice-free day as well, associated with all three sources of climate prediction uncertainties[36]. The scenario uncertainty introduced by the unknown future emissions is two-fold. First, there is a distinct difference between the lowest CMIP6 emission scenario that we analyzed (SSP1-1.9, which reaches a global atmospheric CO$_2$ level of 393 ppm by 2100), which shows no very early ice-free day (earliest projection is 18 years), and the higher CMIP6 emission scenarios (SSP1-2.6 to SSP5-8.5, reaching between 446 ppm and 1135 ppm of atmospheric CO$_2$ concentrations in 2100[37]), which all have members with ice-free days within a decade. The finding that early ice-free days occur under all but the lowest emission scenario considered by CMIP6 (under SSP1-2.6 to SSP5-8.5) without any influence of the strength of the forcing scenario agrees with prior work on the timing of the first ice-free month in the Arctic[2,38]. The second effect of the scenario uncertainty is that under the two lowest emission scenarios (SSP1-1.9 and SSP1-2.6), there is a chance to avoid daily ice-free conditions all together. As SSP1-1.9 tends to stay below 1.5°C of global warming by 2100[19] and SSP1-2.6 is expected to stay around 1.5 °C[19], the possibility to avoid daily ice-free conditions under these low emission scenarios matches with studies that found that for a global warming below the 1.5 °C Paris target[32] any occurrence of a monthly mean ice-free Arctic may still be avoidable[33–35].

Internal variability prediction uncertainty, ranging from 26 years for SSP5-8.5 to more than 58 years for SSP1-2.6 (based on the MPI-ESM1-2-LR model, which provides the largest ensemble of daily data, Supplementary Table S5), also contributes significantly to the overall prediction uncertainty. Thus, while the first ice-free Arctic could occur within 3 years from a daily SIA minimum of 3.39 million km², it may also not occur for over 30 years, due to internal variability. A similarly large internal variability uncertainty has also been found for projections of the first ice-free month[14,39]. This highlights that internal variability uncertainty affects all projections of an ice-free Arctic, be it monthly or daily, limiting the prediction accuracy to a range of at least two decades[14], if not more. In addition, despite performing model selection based on the historical SIA simulation and aligning all projections on the last time a simulation had a September SIA minimum above or at observed 2023 levels, model differences add to the prediction uncertainty, as also seen for monthly ice-free projections[13].

To understand the rare but high-impact possibility of a rapid loss of SIA to 1 millon km² in the daily data from 2023 equivalent conditions, we investigated the storylines of the nine fastest cases, which reached ice-free conditions within 3–6 years. Most of the first ice-free days occur in August, and the first ice-free period lasted between 11 and 53 days. What they all had in common was that the first ice-free day occurred during a Rapid Ice Loss Event (RILE, Fig. 2). What is noteworthy is that for all quick transition cases the 2023 equivalent year occurs after a period of little or no trend in the daily and monthly SIA over the previous 10–15 years, with previously lower SIA than the the 2023 equivalent. This is not dis-similar to the observed SIA evolution in the 15 years prior to 2023 (see Fig. 2). Furthermore, in an investigation of RILE events in CMIP6 models, it has been shown that the probability of a RILE increases by 20% compared to the overall RILE probability after a 10-year stable period in the sea ice extent[26]. These results suggest that if a RILE were to occur in the near future, it could potentially bring us a first ice-free day relatively quickly. However, while the first ice-free day has high symbolic significance, it does not mean that after that the Arctic Ocean becomes ice-free every year. Furthermore, even consistently ice-free conditions in the monthly September mean do not imply ice-free conditions all year, as sea ice continues to return during the dark and hence cold Arctic fall and winter in model simulations until atmospheric CO$_2$ levels exceed 1900 ppm[40].

The primary trigger of the rapid transition to the first ice-free day within 3–6 years that we identified was a warm atmosphere in the previous winter and spring, leading to a loss of sea ice mass year-round (Fig. 3b). The first year with an ice-free day had spring daily mean temperatures already in January, thanks to heatwaves/blockings and/or warm air intrusions (Fig. 4). In addition, we frequently found storms going across the Arctic in the days leading up to the first ice-free day. All these events are projected to increase in frequency as the Arctic warms[30], making the first ice-free day increasingly more likely. The good news is, for all storyline cases, the first ice-free day occurs in years with a 5-year running mean global temperature at or above 1.5 °C compared to pre-industrial level (Table 1). This means that if we could keep warming below the Paris Agreement target of 1.5 °C of global warming[32], ice-free days could potentially still be avoided.

## Methods

### Data and definitions

We analyzed simulations from all models that participated in the Climate Model Intercomparison Project phase 6 (CMIP6[15]) that had daily sea ice on the ocean (siconc) or atmosphere grids (siconca) available on any of the Earth System Grid Federation (ESGF) portals in late May 2024, as well as files that had been previously downloaded onto the Levante server of the German Climate Computing Center (DKRZ). We also obtained their grid cell area (areacello and areacella, respectively). All available ensemble members were used. We used the historical scenario for our model selection (see next subsection), and the Shared Socioeconomic Pathways SSP1-1.9, SSP1-2.6, SSP2-4.5, SSP3-7.0 and SSP5-8.5[16] for our ice-free projections. For the subset of cases that we investigated further from section "Storylines: From 2023 equivalent conditions to the first ice-free day in 3–6 years" onwards, we also used their daily surface air temperature (tas), daily sea level pressure (psl), and their monthly sea ice mass (simass) (monthly due to the unavailability of daily simass in some of these models).

The SIA, SIE and sea ice mass (SIMASS) were calculated north of 30°N, on the model's native grid. SIA was defined as the sum over all grid cells n of the sea ice concentration multiplied by the grid cell area:

$$SIA = \sum_n \mathrm{siconc}(n) \times \mathrm{areacello}(n). \tag{1}$$

SIE was defined as the sum of the grid cell area for all grid cells m where the sea ice concentration was larger than 0.15:

$$SIE = \sum_m \text{areacello}(m). \tag{2}$$

As recommended in previous studies[2,41], we conducted our analyses using SIA, with the exception of the RILE analysis, which uses SIE for consistency with prior RILE studies[23,25].

The first ice-free year was defined as the first year where daily or monthly SIA is lower than or equal to 1 million $km^2$.

RILEs were defined based on the September monthly SIE, following[25], which means that a RILE is defined as a period of at least 4 years for which the trend in the 5-year running mean minimum SIE is lower than $-0.3$ million $km^2$/year.

The warming of the models compared to pre-industrial was computed by using the daily surface air temperature for the first 50 years of the pre-industrial control simulation (on ensemble member r1i1p1f1); taking the area-weighted global temperature, averaged over these 50 years; and subtracting it from the area-weighted global temperature averaged 2 year prior to the first year with an ice-free day until 2 years after that first year (i.e., over a 5 year period).

All analyses using daily data were performed on a no-leap 365 day calendar, and models that produced output on a standard calendar with leap years had their Feb 29th data dropped. Models that used a 360 day calendar (specifically UKESM1-0-LL and HadGEM3-GC31-LL) were not included in the analysis, as their results can not be directly compared with models with 365 days when analyzing the timing of daily ice-free conditions.

To determine whether the year when ice-free conditions were reached was anomalous, we computed the anomalies between that final year and all years since 2023-equivalent (see below), for the Central Arctic Ocean north of 80°N and all neighboring shelf seas, in the 10th and 90th sea level pressure percentiles, and in heating degree days (HDD), for each season. HDDs were calculated with 0 °C as a reference; using e.g. the freezing temperature of sea water instead did not significantly modify our results. The seasons have different numbers of days (indicated in the caption of the corresponding supplementary table), to reflect the Arctic seasonal cycle more accurately. Using three-month seasons instead did not significantly change our results.

### Model selection
Due to the large model spread in simulations of Arctic sea ice evolution[2], we used two SIA based criteria to select the models that performed best in the historical CMIP6 simulations.

The first criterion used here was that the simulated September monthly mean falls within the satellite-derived SIA in September over the 20-year period spanning 1995–2014, plus/minus the average standard deviation of the 1995–2014 average September SIA in models with more than 5 members ($\pm 0.38$ million $km^2$). To fully account for the influence of internal variability, we only excluded models if all available ensemble members failed to meet the criterion for inclusion. To account for observational uncertainty, we used the monthly SIA calculated from daily NOAA/NSIDC CDR SIC data, version 4[17], using the Bootstrap SIA[42] as upper bound and the NASA Team SIA[43] as lower bound (both from the[17] dataset, which fills the pole hole). For September, the 1995–2014 mean difference between the two is $-1.16$ million $km^2$. This criterion means that we exclude models that have a mean state over the last 20 years of the historical simulations that is too high or low compared to observations. Note that changing the evaluation period has a small effect on which models are selected. However, the details of the evaluation period do not affect the overall results presented here, beside the exact models and number of simulations included in the analysis.

The second criterion used here was that the day of the minimum daily SIA from the simulations between 1980 and 2014 falls within the observational range of day 238 (August 26, from the CDR SIA data[17]) to day 272 (Sept 29, from the NASA Team SIA data[17]), plus/minus 5 days (the average standard deviation of the sea ice minimum day over 1980–2014 from the models with more than 5 members). As for the first criterion, if any ensemble member from a model fell within the observational range, the criterion was considered to be met. We chose this second criterion to ensure that models simulate the daily minimum at a time comparable to observations, since the focus of this study is on the first ice-free day.

Both criteria had to be met for a model to be retained. Applying these two criteria reduced the number of models from 29 to 11 models, with criterion 1 excluding the most models. The 11 models retained are listed and cited in Supplementary Table S5, including the number of ensemble members per model and simulation. Across these 11 models, this includes a total of 366 scenario simulations, spanning SSP1-1.9 to SSP5-8.5, with 34 for SSP1-1.9, 86 for SSP1-2.6, 89 for SSP2-4.5, 86 for SSP3-7.0 and 71 for SSP5-8.5. As some models provided many more ensemble members than others (e.g., the MPI-ESM1-2-LR providing 205 of the 366 total scenario simulations), we used the earliest and latest ensemble member from each model and scenario in Fig. 1, to avoid biasing the multi-model analysis towards models with more ensemble members.

### Detection of the 2023 equivalent year
In order to assess the storyline of how soon the Arctic Ocean could be ice-free (1 million $km^2$ or less of SIA remaining), we start our analysis from the last year the daily SIA minimum was equal to or larger than the observed 2023 daily SIA minimum before a simulation reaches daily ice-free conditions for the first time. The observed 2023 daily SIA minimum was 3.39 million $km^2$, based on the daily SIA calculated from the pole-hole filled daily NOAA/NSIDC CDR SIC data, version 4[17].

Some ensemble members from the selected models had to be discarded because their 2023 equivalent pre-dated the beginning of the scenario simulations, as their daily SIA minimum was below 3.39 million $km^2$ at the beginning of the scenario simulations, but their historical simulation could not be obtained (broken link, corrupted data, not turning up on any of the portals).

### Data availability
The CMIP6 data are freely available on the Earth System Grid Federation (ESGF, https://esgf-node.llnl.gov/search/cmip6/, https://esgf-metagrid.cloud.dkrz.de/search, https://esgf-node.ipsl.upmc.fr/projects/cmip6-ipsl/ and https://esgf-ui.ceda.ac.uk/cog/search/cmip6-ceda/). The NOAA/NSIDC Climate Data Record of Passive Microwave Sea Ice Concentration, Version 4[17] are freely available at https://nsidc.org/data/g02202/versions/4. The derived daily SIE and SIA data are publicly available on the Arctic Data Center at https://doi.org/10.18739/A2CC0TV9V[44].

### Code availability
All codes used for this work are publicly available on Zenodo at https://doi.org/10.5281/zenodo.14006059.

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

## Acknowledgements

The authors acknowledge funding from the Swedish National Research Council Starting Grant award 2018-03859 (CH), the Swedish National Space Agency award 2022-00149 (CH), and the NSF-CAREER award 1847398 (AJ). We acknowledge the World Climate Research Programme, which, through its Working Group on Coupled Modelling, coordinated and promoted CMIP6. We thank the Sea Ice Model Intercomparison Project (SIMIP) for requesting daily sea ice output for CMIP6. We thank the climate modeling groups for producing and making available their model output, the Earth System Grid Federation (ESGF) for archiving the data and providing access, and the multiple funding agencies who support CMIP6 and ESGF. CH acknowledges the data access and computing support provided by the Deutsches Klimarechenzentrum (DKRZ) fourth High Performance Computer System for

Earth System Research (HLRE-4) Levante. AJ acknowledges the data access and computing support provided by the NCAR CMIP Analysis Platform (https://doi.org/10.5065/D60R9MSP) as well as computing support from the Casper system (https://ncar.pub/casper) provided by the NSF National Center for Atmospheric Research (NCAR), sponsored by the National Science Foundation.

## Author contributions

C.H. and A.J. jointly and with equal contributions conceptualized the article, obtained and analyzed data, produced figures and wrote the article.

## Funding

## Competing interests

The authors declare no competing interests.
