## [Transparent Peer Review file · Nature Communications]

The first ice-free day in the Arctic Ocean could occur before 2030

Corresponding Author: Dr Céline Heuzé

Version 0:

Reviewer comments:

Reviewer #1

(Remarks to the Author)

The manuscript analyzes the occurrence of the first day when the Arctic becomes essentially ice-free based on CMIP6 climate models. As the authors point out, we will observe an ice-free Arctic in the daily data first before seeing it in the monthly data (which is usually analyzed for these purposes). Since such an event would create a lot of attention and mark the beginning of a seasonally ice-free Arctic with all its implications, it is relevant and of high impact to estimate when this could be the case. The authors find a broad range of timing for the first ice-free day, subject to model and scenario uncertainty, and internal variability, arguably the largest near-term source of uncertainty. In the second part of the manuscript, the authors examine the interesting, although somewhat unlikely, case of the first ice-free day happening within 3-6 year from 2023, i.e., before 2030. By analyzing the simulations that include such an event, they shed light on how it might happen, and what might trigger it, and find winter and spring preconditioning, and warm and stormy conditions as a common feature.

Overall, the manuscript is easy to follow, and the figures are clear. The manuscript offers novel insights by explicitly considering the first daily ice-free occurrence, something which has not been investigated very often. However, I think the analysis falls short on answering the question what might trigger a rare but possible quick transition to ice-free conditions before 2030, and could be improved.

The authors show characteristics of these fast ice-free events, showing enhanced warming and storminess leading up to the first ice-free day. But what really makes these events special so that they lose the ice so quickly within a few years? What distinguishes a simulation that loses the ice in 5 years from one that loses it in 15 years? The analysis could focus more on comparing the trends in these simulations with observed or simulated trends in other periods, to tease out what's special.

Similarly, the authors do a nice job in distinguishing between preconditioning in the prior years before the event, and 'final-year triggers'. However, the analysis of the final year triggers partly misses the comparison to other future years (and not just with the climatology). Are the final years more stormy than normal, or in any other way unusual compared to the other simulations? The authors should include a more rigorous analysis of these last years, as right now, section 2.2.2 seems a bit anecdotal to me.

Finally, the authors state in the title that there is a 7% chance of such a rapid event leading to ice-free conditions before 2030. I don't think this number represents the real probability, since it's just derived from counting CMIP6 simulations which is dependent to model selection, ensemble size for the selected models, etc. Furthermore, as the authors mention, all the fast ice-free simulations show the same slowdown in sea-ice loss over the last 10-15 years which is also seen in observations. Would that not increase the probability of it happening in reality? I would suggest to removing the number and instead mention a 'small chance'.

Minor comments:

The authors should provide information on the ensemble size of each selected model and scenario.

Table 1: It would be interesting to show the percentage of members showing rapid ice-free conditions for each of the 4 models, either here or in the Extended Data Tables 1 and 2.

L104 – 105: Is this surprising given the selection? Does it reveal something about the importance of preconditioning?

L126 – 137: Is this something which is specific to the quick transition simulations, or is that similar in the slower transitions to ice-free conditions?

Extended Data Figure 3: Legend says 'Pre-2023 mean pressure', should be temperature.

Reviewer #2

(Remarks to the Author)

The paper presents analysis of daily sea ice data from CMIP6 scenario simulations to establish the timing of the first ice-free day in the Arctic with further investigation of the driving mechanisms behind the low-probability scenario of a very rapid transition to an ice-free day from current sea ice area conditions. The focus on the first ice-free day is novel and the approach is well established and thoroughly documented. While it is refreshing to see a new analysis of the timing of ice-free conditions in the Arctic I think the paper needs some major revisions prior to publication.

Major comments:

The authors hinge the motivation of the paper on the importance of ice-free conditions in the Arctic on climate, economics, ecosystems and geopolitics, but there are two issues with this: (1) this threshold of 1 million square kilometers is arbitrary and not directly related to known tipping elements in any of these fields, so it is the continuum of change which matters. (2) if it was just 1 ice-free day would it matter for climate if the ice 'bounced back' the next day? So I think it'd be better to focus the story on the RILE(s) that lead to this threshold being crossed as it is the pace of change which is critical for system adaption in all fields i.e. the high-impact part of this storyline approach is not that the million square kilometre threshold is passed but rather that we get a RILE that causes an extremely rapid drop in SIA such that this threshold is crossed with just a few years. This is a choice of framing of the results rather than a critique of the results themselves, but I think it is more valuable for readers across fields to emphasise the possibility of a sudden collapse of SIA in just a few years rather than the expectation of 2-3 decades, rather than putting too much emphasis on a particular threshold.

I think you need some more discussion about the extent to which observations support or not the possibility of such a RILE occurring. If you take the maximum observed drop in SIA from one year to the next, and you had this every year, would that be enough to reach an ice-free Arctic within 6 years? If the kind of RILEs seen in these extreme cases in the models have never even occurred in the observations in a single year, how much confidence can we have that this is anything other than a model-world issue?

For example, if you regress annual-minimum daily SIA against cumulative CO2 emissions (a la Notz and Stroeve 2016) then you keep an expectation value of daily SIA crossing the million sq km threshold around 2060 with no probability of it reaching this threshold by 2030.

Also given that you introduce the concept of focusing on the daily SIA data by saying the first-ice free day will be seen in the daily satellite data I think a bit more inclusion of observational data is needed. I know you mention the inconsistency between observed and modelled RILEs (line 112), but this is a major issue for interpreting all this model analysis so I think it really warrants further investigation.

These models you've selected for the storyline: where do they sit in the winter negative feedback space? Do they manage to get such RILEs because they have a greatly under-estimated negative feedback in response to summer ice loss?

Regarding the transition to an ice-free day (Lines 127 onwards): since these models get between 11 and 53 ice free days in the first year that they hit ice-free conditions do you not also always get an ice-free month if a month defined as 30-day running mean rather than following (arbitrary) calendar month definitions?

Model selection is a really challenging issue, but your choice of such a short evaluation period (just 14 years) is a problem given the large role of natural variability on such short timescales (1/3rd of sea ice loss can be due to natural variability on such timescales, see e.g. Dorr et al 2023). Are you not just selecting models with the right timing of internal variability?

Reviewer #3

(Remarks to the Author)

The authors use well-defined criteria to choose models from the CMIP6 collection of climate model simulations to analyze when the first day of "ice-free" (area less than $1.0 \times 10^6 \text{ km}^2$) conditions might occur in the Arctic Ocean. Their results are similar to those found for the first ice-free month in the literature, which is cited. They analyze the conditions driving the first ice-free events, the results of which are interesting and explained well. The methodology and data analyses are sound, and the methods section is suitably detailed.

The authors include a wide range of climate scenarios in the analyzed simulation set, which provide insight into the physical system and forcing driving the events. However, their figures tend to emphasize individual model results rather than the scenarios. I believe the paper's message would be stronger if the individual model results were de-emphasized. Partly because of the model detail, the conclusions that should be drawn from the figures are often not immediately obvious without

reading the text. Simplifying them and adding more explicit pointers to what should be seen would be helpful.

One worry: It is challenging to publish these kinds of results, particularly with a “headline” like “There is a N% risk that ... could occur before 2030”, without being accused of fear-mongering. Climate change is real, and it's really scary, but I do think that a lot of people don't really understand sea ice or polar systems more generally. The authors have done a good job of pointing out the variability inherent in the system, but I still think it would be helpful to be explicit about what polar change means for sea ice over the next decades. In particular, winter is white (but the ice is thinner/weaker). The present results do indicate that the first ice-free day doesn't mean it will become the norm (i.e. every year), but the authors could point that out explicitly. The first ice-free day is symbolic (because it's defined based on an arbitrary limit of $1.e6 \text{ km}^2$), and it is almost certainly a harbinger of things to come, but it does not necessarily have immediate, far-reaching consequences, even if it's the first day of several. This type of context would help the general reader understand the consequences and what to expect without jumping to conclusions based on the title. My personal opinion is that “7%” in the title is more likely to produce a “meh” response for those who aren't familiar with the very real and enormous consequences of sea ice loss in the Earth system. So, my recommendation is to think a bit more about how these results are framed.

Section 2.1

It would be helpful to explain in broad terms what the SSP levels mean, for the general reader.

What exactly is figure 1 supposed to illustrate? Is it just documenting the models and SSP scenarios?

Is the latest ice-free day (shown in figure 1) discussed in detail? If not, put this in the supplementary material.

Section 2.2

Fig 3 is difficult to quickly interpret and it is not convincing to me. Is the conclusion true if you average over the models, to simplify the figure? If not then I think you have other issues to address!

Regarding “Final-year triggers”, it would help to discuss the average number of years of preconditioning in the runs. E.g. place the first grey dashed line in context of the purple bars in figure 2, explicitly.

Figure 4

Explain what the horizontal lines are, in the caption.

Could this figure be simplified? Are the colors important? At the very least, use the same color for the same SSP conditions, and I suggest not separating out the various models and versions.

What does “normal” look like?

Conclusions

Clarify what the SSP levels mean in everyday terms. The discussion assumes the reader is familiar with them. If it's still acceptable among the climate research community, it might be helpful to label them (e.g. ‘Paris target’, ‘business as usual’ or similar) instead of using the complicated numbering system — otherwise explain what those numbers mean. Relate them to current/expected conditions, if possible.

The abstract and introduction gave some general motivation for this study, but the conclusions left me asking “why do we care?” The Paris Agreement target of 1.5C is somewhat arbitrary, chosen mainly for galvanizing political support. With regards to impacts of a daily sea ice area level of $1.e6 \text{ km}^2$ and the physical/ecological system motivations mentioned in the introduction, why is this particular number (1.5C) scientifically significant?

Extended Data Fig 1 - red and pink are nearly indistinguishable

Extended Data Fig 3, 4 - please put the take-away message for the figures in the captions

Line 75 use “less” instead of “lower”

Line 80 use “among” instead of “between”

Line 121 isn't the expression “year-round” instead of “year-around”?

Line 166-168 fix sentence grammar. Perhaps change “on” to “in”, “compared to” to “above the”, and put a comma after “level”.

Line 170 change “stayed” to “stays” and remove “less than”

Check grammar in the figure captions.

Version 1:

Reviewer comments:

Reviewer #1

(Remarks to the Author)

I thank the authors for carefully considering and addressing the comments, and I agree that the analysis is now more rigorous in establishing what factors would contribute to the unlikely but possible scenario of an ice-free day within 6 years. I like that the authors now compare the ice loss in the fast ice loss simulations to observed short-term variability, indicating we could have already produced a RILE under the right circumstances.

I have a few minor comments:

Cooling degree days: The authors might want to check what the correct unit for cooling degree days is. I am not completely sure, but I thought it was cooling degree days, not °C.

L. 144-147: I like this part where the authors indicate that a RILE could have already happened in the observed record. I think this sentence is a bit wordy though, and could be shortened to make the message more clear.

Reviewer #2

(Remarks to the Author)

I'm very satisfied with how you have responded to my comments, thank you for the nice work.

There was a a point where I think we had a misunderstanding as I hadn't explained myself well:

For example, if you regress annual-minimum daily SIA against cumulative CO2 emissions (a la Notz and Strove 2016) then you keep an expectation value of daily SIA crossing the million sq km threshold around 2060 with no probability of it reaching this threshold by 2030.

That approach would indeed get us a first ice-free day in the mid 21st century, but that would be the average expected time when we would, due to capturing the forced response (as in Notz and Strove 2016, the focus is on the relationship between the mean SIA and the cumulative emissions). It would not get us the earliest possible dates (the early outer edge of the probability distribution around the average, >2 standard deviations around the mean). As you noted earlier in your comments, it is the rapid nature of the possible transition to ice-free conditions that is important to emphasize, compared to the most likely (=middle of the distribution) timing. Which is why we focus on the quick transition members.

But what I had done was bootstrap the probability of an ice-free day by 2030 based on the observed relationship between annual-minimum SIA and log(CO2), using an AR1 model to simulate natural variability based on observed variability around the trend. In none of the 10,000 members of the bootstrap did I find SIA<1 million sq km by 2030. Hence I questioned whether such RILEs are really feasible. However, I think you discussed very nicely in the text why one may not have observed evidence of such a RILE yet, so I don't see a need for further changes.

Reviewer #3

(Remarks to the Author)

The authors have addressed my concerns in their revised manuscript and review response.

REVIEWER COMMENTS

The contribution of the reviewers has been duly acknowledged, in the acknowledgement section:

“Finally, we thank the Editor and the three anonymous reviewers for their suggestions that improved the robustness and clarity of this manuscript.”

In this document, the reviewers’ comments are in black, and our response in blue.

The comments are addressed sequentially, with Reviewer 1 pages 1-4, Reviewer 2 pages 5-8, and Reviewer 3 pages 9-13.

Reviewer #1 (Remarks to the Author):

The manuscript analyzes the occurrence of the first day when the Arctic becomes essentially ice-free based on CMIP6 climate models. As the authors point out, we will observe an ice-free Arctic in the daily data first before seeing it in the monthly data (which is usually analyzed for these purposes). Since such an event would create a lot of attention and mark the beginning of a seasonally ice-free Arctic with all its implications, it is relevant and of high impact to estimate when this could be the case. The authors find a broad range of timing for the first ice-free day, subject to model and scenario uncertainty, and internal variability, arguably the largest near-term source of uncertainty. In the second part of the manuscript, the authors examine the interesting, although somewhat unlikely, case of the first ice-free day happening within 3-6 year from 2023, i.e., before 2030. By analyzing the simulations that include such an event, they shed light on how it might happen, and what might trigger it, and find winter and spring preconditioning, and warm and stormy conditions as a common feature.

Overall, the manuscript is easy to follow, and the figures are clear. The manuscript offers novel insights by explicitly considering the first daily ice-free occurrence, something which has not been investigated very often.

We thank the reviewer for their time spent reviewing the manuscript. Their questions and comments have helped us improve the rigor of the analysis.

However, I think the analysis falls short on answering the question what might trigger a rare but possible quick transition to ice-free conditions before 2030, and could be improved. The authors show characteristics of these fast ice-free events, showing enhanced warming and storminess leading up to the first ice-free day. But what really makes these events special so that they lose the ice so quickly within a few years? What distinguishes a simulation that loses the ice in 5 years from one that loses it in 15 years? The analysis could focus more on comparing the trends in these simulations with observed or simulated trends in other periods, to tease out what’s special.

A comparison of the simulated trends in the quick transition simulations with observations is a very good idea. We have now added discussion of how the sea ice loss in the quick transition simulations and in a RILE event in general compares to observed sea ice losses to date. Specifically, we now discuss that we would only need three sea ice loss events similar to 2011 to 2012 to get from 2023 equivalent conditions to ice-free conditions (and less than two 2006-2007 ice-loss events in a row). Furthermore, we added that while the 2004-2007 observed 5yr running mean SIE trend fell 0.08 million km²/year short of qualifying as a RILE, two extreme September SIE loss events such as 2006-2007 and 2011-2012 within a 4yr period would have led to a RILE.

Similarly, the authors do a nice job in distinguishing between preconditioning in the prior years before the event, and ‘final-year triggers’. However, the analysis of the final year triggers partly misses the comparison to other future years (and not just with the climatology). Are the final years more stormy than normal, or in any other way unusual compared to the other simulations? The authors should include a more rigorous analysis of these last years, as right now, section 2.2.2 seems a bit anecdotal to me.

We thank the reviewer for these suggestions. We have now added a more rigorous analysis of the atmospheric conditions of the final year, in the central Arctic Ocean and the peripheral seas where the storms come from, comparing their statistics to those of the years prior. The final year is indeed anomalous; see the new Extended Data Table 3 and corresponding additions in the text.

Finally, the authors state in the title that there is a 7% chance of such a rapid event leading to ice-free conditions before 2030. I don’t think this number represents the real probability, since it’s just derived from counting CMIP6 simulations which is dependent to model selection, ensemble size for the selected models, etc. Furthermore, as the authors mention, all the fast ice-free simulations show the same slowdown in sea-ice loss over the last 10-15 years which is also seen in observations. Would that not increase the probability of it happening in reality? I would suggest to removing the number and instead mention a ‘small chance’.

We thank the Reviewer for this comment. We agree that the 7% depend on the available set of simulations, which SSPs are included in the analysis, etc, and as such indeed most likely do not represent the actual probability of occurrence. The main point we wanted to make is that a quick transition to ice-free conditions is a small-probability event, but does occur in some models. In terms of the second point, we do indeed think that the observed slowdown matching the quick transition simulations does suggest an increased probability of a rapid transition. So another reason to not give specific numbers. Thus, we have now removed the specific # in regards to the probabilities of occurrence from the manuscript and title.

The title now reads: The first ice-free day in the Arctic Ocean could occur before 2030

Minor comments:

The authors should provide information on the ensemble size of each selected model and scenario.

This information has now been added to the Supplementary Table 1, as that is the best place to provide this information as the ensemble size differs between scenarios within one model.

Table 1: It would be interesting to show the percentage of members showing rapid ice-free conditions for each of the 4 models, either here or in the Extended Data Tables 1 and 2.

Thank you for this suggestion. We have added that to section 2.2, at the very end right before 2.2.1 starts, as it didn't fit well into the table. The added text reads: "Specifically, only 1% of the MPI-ESM1-2-LR simulations (2 out of 205) are quick transition simulations, 3% of the CanESM5 simulations (2 out of 72), 5% of the ACCESS-CM2 simulations (2 out of 39), and 30% of the EC-Earth3 simulations (3 out of 10; see Supplementary Table S1)"

L104 – 105: Is this surprising given the selection? Does it reveal something about the importance of preconditioning?"

This refers to "Notably, all quick transition simulations meet the criteria for a Rapid Ice Loss Event (RILE) during September".

If you are referring to the quick transition members as "the selection", then yes, it is not surprising that the ice-loss is quick, as that is indeed how these are identified, by choosing the models that transition from 2023 conditions to ice-free for at least one day within 3-6 years. But this selection of the quick transition members from all available members does not take preconditioning into account, as all simulations are assessed in terms of how long it takes from 2023 equivalent conditions to the first ice-free day. The fact that the simulations that transition to ice-free most quickly indeed meet the existing requirement for a RILE event for September and basically for August in all of the 9 quick transition simulations, however, is not baked into the selection, and thus is noteworthy. If you are referring to the overall model selection with "the selection", then the answer is that this does not affect the models that simulate the quickest conditions, as we select not individual ensemble members based on their fit to the observations over now 1995-2014 (and 2000-2014 before), but include all models where at least one ensemble member fits the observations. So again the selection does not reveal anything about pre-conditioning for the specific ensemble members that show quick transitions.

Since we are not sure what exactly the question was here, we were not able to make changes to the manuscript directly based on this comment. But in response to reviewer 3, we have made it clearer that we retain all models that have at least one ensemble member that falls within the observations plus observational uncertainty, rather than selecting individual ensemble members.

L126 – 137: Is this something which is specific to the quick transition simulations, or is that similar in the slower transitions to ice-free conditions?

Refers to the statement that they do not just have ice-free day but at least 11.

This is a good point. In all CMIP6 simulations analyzed, 4 members only have a single ice-free day (spanning all SSPs), with a mean of 25 days and a max of 71 days. We have now added information on the length of the first ice-free period in the daily data to section 2.1, before the discussion of the offset between the first ice-free day and the first ice-free month. So that when the reader gets to the discussion of the ice-free period for the quick transition members, the overall ice-free duration for the CMIP6 models will have already been established.

Extended Data Figure 3: Legend says 'Pre-2023 mean pressure', should be temperature. Corrected.

Reviewer #2 (Remarks to the Author):

The paper presents analysis of daily sea ice data from CMIP6 scenario simulations to establish the timing of the first ice-free day in the Arctic with further investigation of the driving mechanisms behind the low-probability scenario of a very rapid transition transition to an ice-free day from current sea ice area conditions. The focus on the first ice-free day is novel and the approach is well established and thoroughly documented. While it is refreshing to see a new analysis of the timing of ice-free conditions in the Arctic I think the paper needs some major revisions prior to publication.

We thank the reviewer for their helpful and constructive comments, which helped us refine the clarity of the manuscript and make it accessible to a wider audience.

Major comments:

The authors hinge the motivation of the paper on the importance of ice-free conditions in the Arctic on climate, economics, ecosystems and geopolitics, but there are two issues with this: (1) this threshold of 1 million square kilometers is arbitrary and not directly related to known tipping elements in any of these fields, so it is the continuum of change which matters. (2) if it was just 1 ice-free day would it matter for climate if the ice 'bounced back' the next day? So I think it'd be better to focus the story on the RILE(s) that lead to this threshold being crossed as it is the pace of change which is critical for system adaption in all fields i.e. the high-impact part of this storyline approach is not that the million square kilometre threshold is passed but rather that we get a RILE that causes an extremely rapid drop in SIA such that this threshold is crossed with just a few years. This is a choice of framing of the results rather than a critique of the results themselves, but I think it is more valuable for readers across fields to emphasise the possibility of a sudden collapse of SIA in just a few years rather than the expectation of 2-3 decades, rather than putting too much emphasis on a particular threshold.

Thank you for this suggestion. We have incorporated this into the revised manuscript in multiple ways:

In the Abstract: We have removed the ecological and other implications of an ice-free Arctic, and instead describe why an ice-free Arctic is such an important symbolic threshold. It now reads “The transition from an white, ice-covered Arctic Ocean to a blue Arctic Ocean without sea ice is a highly symbolic event as it visibly demonstrates the impact of anthropogenic greenhouse gas emissions on the climate system. “

In the Introduction: We have rewritten this part to now distinguish between the symbolic significance of even a single day of ice-free conditions and the implications of a permanent transition to ice-free conditions in the summer. “While the first occurrence of ice-free conditions has primarily symbolic significance, a transition to an Arctic Ocean that regularly has a sea ice area of less than 1 million km² (commonly used as the ice-free threshold) in the summer is expected to have cascading effects on the rest of the climate system”

At the end of the introduction, we now also added “the goal is to raise awareness for the potential of a rapid loss of sea ice in the near future,” to emphasize the possibility of a sudden collapse of SIA.

I think you need some more discussion about the extent to which observations support or not the possibility of such a RILE occurring. If you take the maximum observed drop in SIA from one year to the next, and you had this every year, would that be enough to reach an ice-free Arctic within 6 years? If the kind of RILEs seen in these extreme cases in the models have never even occurred in the observations in a single year, how much confidence can we have that this is anything other than a model-world issue?

This is a valid point. We have done some more analysis based on the NSIDC CDR SIE record, and based on that, have added the following to the discussion about RILEs (around line 112 in the initial manuscript).

“RILEs have been found in all CMIP6 models and all months of the year (Sticker et al 2024), and as their name implies, describe a rapid loss of sea ice over a short period of time. The sea ice extent loss during a RILE exceeds even the September SIA loss observed in the early 21st century (Sticker et al 2024), even though the 4yr period between 2005 and 2007 got close, with an average trend in the 5-year running mean sea ice extent in September reaching -0.22 million km² (compared to the 0.3 million km² threshold for a RILE). Had we seen a second extreme SIE drop in September as observed between 2006 and 2007 (of 1.5 million km²) or even one as seen between 2011 and 2012 (-0.99 million km²) within the 4yr period around 2007, that period would have qualified as a RILE. Given that RILEs in CMIP6 models are most likely to start when the Arctic September SIE is at slightly less than 4 million km² (Sticker et al 2024), it is not unexpected that we have not seen a RILE yet. But given the observed record sea ice loss events of 2007 and 2012, RILEs seem entirely possible.”

For example, if you regress annual-minimum daily SIA against cumulative CO₂ emissions (a la Notz and Strove 2016) then you keep an expectation value of daily SIA crossing the million sq km threshold around 2060 with no probability of it reaching this threshold by 2030.

That approach would indeed get us a first ice-free day in the mid 21st century, but that would be the average expected time when we would, due to capturing the forced response (as in Notz and Strove 2016, the focus is on the relationship between the mean SIA and the cumulative emissions). It would not get us the earliest possible dates (the early outer edge of the probability distribution around the average, >2 standard deviations around the mean). As you noted earlier in your comments, it is the rapid nature of the possible transition to ice-free conditions that is important to emphasize, compared to the most likely (=middle of the distribution) timing. Which is why we focus on the quick transition members.

Also given that you introduce the concept of focusing on the daily SIA data by saying the first-ice free day will be seen in the daily satellite data I think a bit more inclusion of observational data is needed. I know you mention the inconsistency between observed and modeled RILEs

(line 112), but this is a major issue for interpreting all this model analysis so I think it really warrants further investigation.

As described above, we have now performed additional analysis that shows that RILE events are entirely possible based on observed sea ice extent losses, they just haven't yet occurred based on the specific sequence of events we have seen, and it is not unexpected that they have not occurred as the peak of the probability distribution in terms of the mean September sea ice extent based on the models is still a bit lower than the current mean September sea ice extent.

These models you've selected for the storyline: where do they sit in the winter negative feedback space? Do they manage to get such RILEs because they have a greatly underestimated negative feedback in response to summer ice loss?

The quick transition simulations, from 4 different CMIP6 models, do not have especially strong or weak positive or negative feedbacks or model physics that would make them more likely to show rapid sea ice loss (RILE) events. We know that because other ensemble members from the same four models take much longer to reach ice-free conditions. We tried to explain this in lines 99-101. Furthermore, RILEs occur for almost all CMIP6 models, as shown in Sticker et al., (2024). This is cited in section 2.2.1. This means that it is not these four models that have some special features, but rather the internal variability in specific ensemble members that lead to the quick transitions. We have now expanded this explanation at the end of section 2.2. (right before 2.2.1 starts), to make this clearer to future readers.

Regarding the transition to an ice-free day (Lines 127 onwards): since these models get between 11 and 53 ice free days in the first year that they hit ice-free conditions do you not also always get an ice-free month if a month defined as 30-day running mean rather than following (arbitrary) calendar month definitions?

By definition, yes, the simulations that have close to 30 or more than 30 ice-free days per year would have an ice-free monthly mean if a monthly mean were defined as a 30-day period starting on any day. But monthly mean products, both from climate model output or observations (be it in-situ, reanalyses, or satellite-based) use the calendar months. Therefore, despite sounding absurd, we can have more than 30 ice-free days without having an ice-free month in those products.

Model selection is a really challenging issue, but your choice of such a short evaluation period (just 14 years) is a problem given the large role of natural variability on such short timescales (1/3rd of sea ice loss can be due to natural variability on such timescales, see e.g. Dorr et al 2023). Are you not just selecting models with the right timing of internal variability?

As we include models as long as at least one ensemble member matches the criteria, we do not select for internal variability that matches observations, but exclude models that do not

at all overlap with observations, considering observational uncertainty and internal variability. The fact that “we include models as long as at least one ensemble member matches the criteria” was noted in the methods but not directly in the section where we described the first criteria. We have now added it for both criteria 1 and 2 to make this clear.

Furthermore, we assessed the influence of the model selection period, when we initially decided on the selection and again now. While the exact period used does affect whether a few individual models are included or not, it does not include or exclude the four models with the quick transitions. Nor does it change the results in any other significant way except to slightly change the number of simulations included. As we do agree that a 20yr evaluation period is a better standard to set than a 15yr period, we have now changed the assessment to use the 20yr 1995-2014 period. That leads to the exclusion of two additional models. We now use 1995-2014 rather than an earlier period to focus on the performance in the most recent period, as we want to select models that are not already ice-free or close to ice-free during the end of the historical simulation, and extending the assessment period back too far retains too many models that have a too low SIA in the most recent two decades.

Reviewer #3 (Remarks to the Author):

The authors use well-defined criteria to choose models from the CMIP6 collection of climate model simulations to analyze when the first day of “ice-free” (area less than $1.e6 \text{ km}^2$) conditions might occur in the Arctic Ocean. Their results are similar to those found for the first ice-free month in the literature, which is cited. They analyze the conditions driving the first ice-free events, the results of which are interesting and explained well. The methodology and data analyses are sound, and the methods section is suitably detailed.

We thank the reviewer for their constructive and thoughtful comments on our article, which have helped us better frame the results and make the article accessible to a broader audience.

The authors include a wide range of climate scenarios in the analyzed simulation set, which provide insight into the physical system and forcing driving the events. However, their figures tend to emphasize individual model results rather than the scenarios. I believe the paper’s message would be stronger if the individual model results were de-emphasized.

We have now clarified that the scenarios do not matter for the earliest ice-free days, by adding the text below at the end of the first result paragraph. That is the reason why we emphasize the individual model results, and the internal variability in them that leads to the early ice-free days, rather than the scenario. But we completely agree that this was not clearly stated, and we hope we have clarified that now in the article.

“Notably, the emission scenario does not play an important role here, with the 34 ensemble members that reach ice-free conditions within 10 years from 2023-equivalent conditions occurring under all scenarios except the very lowest emission scenario (SSP1-1.9, Fig. 1a). In fact, the fastest three transitions (in 3 to 4 years) occur under SSP1-2.6, the second lowest CMIP6 forcing scenario. This clearly shows that these rapid transitions from 2023-equivalent conditions to the first ice-free day occur primarily due to internal variability, not due to the strength of the external forcing. The large influence of internal variability on the earliest ice-free days agrees with findings for the earliest first ice-free month (REFS).”

Partly because of the model detail, the conclusions that should be drawn from the figures are often not immediately obvious without reading the text. Simplifying them and adding more explicit pointers to what should be seen would be helpful.

We thank you for the suggestions about the figures. We have now simplified Figure 3 and 4 by showing multi-model averages instead of all quick transition simulations individually, to improve the clarity of the figures. And we have also added statements to all figure captions what should be seen in all figures.

One worry: It is challenging to publish these kinds of results, particularly with a “headline” like “There is a N% risk that ... could occur before 2030”, without being accused of fear-mongering. Climate change is real, and it’s really scary, but I do think that a lot of people

don't really understand sea ice or polar systems more generally. The authors have done a good job of pointing out the variability inherent in the system, but I still think it would be helpful to be explicit about what polar change means for sea ice over the next decades. In particular, winter is white (but the ice is thinner/weaker). The present results do indicate that the first ice-free day doesn't mean it will become the norm (i.e. every year), but the authors could point that out explicitly. The first ice-free day is symbolic (because it's defined based on an arbitrary limit of $1 \times 10^6 \text{ km}^2$), and it is almost certainly a harbinger of things to come, but it does not necessarily have immediate, far-reaching consequences, even if it's the first day of several. This type of context would help the general reader understand the consequences and what to expect without jumping to conclusions based on the title. My personal opinion is that "7%" in the title is more likely to produce a "meh" response for those who aren't familiar with the very real and enormous consequences of sea ice loss in the Earth system. So, my recommendation is to think a bit more about how these results are framed.

We thank the reviewer for these thoughtful comments. We have revised the framing in accordance with their suggestion, and now clearly highlight the difference between crossing the 1 million km^2 threshold for the first time (symbolic) and the climatic, ecosystem, and economic implications of an ice-free Arctic summer year after year (in the Introduction). We have also further clarified in the conclusions that a first ice-free day does not imply ice-free conditions will be the norm year-after-year, nor does it imply the Arctic would be ice-free year around.

"However, while the first ice-free day has high symbolic significance, it does not mean that after that the Arctic Ocean becomes ice-free every year. Furthermore, even consistently ice-free conditions in the monthly September mean do not imply ice-free conditions all year, as sea ice continues to return during the dark and hence cold Arctic fall and winter in model simulations until atmospheric CO_2 levels exceed 1900 ppm (Jahn and Holland, 2013)"

We have also removed the % probability values from the manuscript, following reviewer 1's remarks as well as the changes in how we frame the results.

The revised title is now: The first ice-free day in the Arctic Ocean could occur before 2030

Section 2.1

It would be helpful to explain in broad terms what the SSP levels mean, for the general reader.

Thank you for this suggestion, which is important to make the article useful for a broad audience. We now introduce the SSPs and the meaning of SSP in the introduction. We also now clarify at the end of the first result paragraph that for the earliest ice-free days, the forcing (and hence SSP) does not matter. And we added descriptions of the meaning of the SSPs such as "lowest forcing scenario (SSP1-1.9)" and "second lowest CMIP6 forcing scenario (SSP1-2.6)" as well.

What exactly is figure 1 supposed to illustrate? Is it just documenting the models and SSP scenarios?

We added the following take-home message to the figure caption: This figure shows that the earliest ice-free day does not depend on the forcing scenario but rather on internal variability; but that the stronger the forcing, the shorter the time between the earliest and latest ensemble member reaching the first ice-free day.

Is the latest ice-free day (shown in figure 1) discussed in detail? If not, put this in the supplementary material.

Yes, it is discussed in lines 71-78 in the initial submission. But it was referred to as “slowest” ensemble member, rather than “latest” as in Figure 1. We have now changed all mentions from “slowest” to “latest” to make the connection between the text and the figure 1 clearer. As the forcing impact can be seen primarily in the latest ice-free member, we think it is important to keep the figures showing the latest ice-free members along the earliest ice-free members.

Section 2.2

Fig 3 is difficult to quickly interpret and it is not convincing to me. Is the conclusion true if you average over the models, to simplify the figure? If not then I think you have other issues to address!

As suggested, we have now simplified Figure 3 by averaging across the quick transition simulations. As the 9 different quick transition simulations take a different number of years to transition from the 2023 equivalent year to the first ice-free day (between 3 and 6 years), we now only include the years that are comparable between the simulations (last year at or above 2023 daily min SIA, first year below, last year before the ice-free day, and year of first ice-free day), to allow us to show a multi-simulation average as suggested. Since that reduces the panels from 9 to 1, we have also added a simplified version of the Extended Data Figure 2 as panel 3b, to show both SIA and sea ice mass on the transition to the first ice-free day.

Regarding “Final-year triggers”, it would help to discuss the average number of years of preconditioning in the runs. E.g. place the first grey dashed line in context of the purple bars in figure 2, explicitly.

As we understand this comment, the reviewer is asking how long before the 2023 year the RILE events start. We have calculated that now for all of the quick transition simulations and have added a short discussion of that to the end of the first paragraph of section 2.1.1. In the mean, the September RILES start 1.7 years before the 2023 equivalent year, with a range of 4 years before and in the same year as the 2023 equivalent year, with a median of 1. For the August RILEs, the mean and median is 1 year earlier, but it ranges from 3 years before and 2

years after the 2023 equivalent year. This means the pre-condition varies, and see no relationship between longer pre-conditioning (RILE starting before the 2023 equivalent) and a faster transition to ice-free in the quick transition members.

Figure 4

Explain what the horizontal lines are, in the caption.

Could this figure be simplified? Are the colors important? At the very least, use the same color for the same SSP conditions, and I suggest not separating out the various models and versions.

What does “normal” look like?

As suggested, we have now simplified Figure 4. As we explained in response to one of the Reviewer’s previous comments (and have clarified in the text), the SSPs do not matter for the first ice-free day. The first ice-free day is a result of internal variability. Therefore, we cannot group lines by SSP. Instead we now show the average of all 9 quick simulations on Figure 4 to simplify the figure. This average shows a strong agreement between the simulations regarding the persistent atmospheric events (heatwaves and blocking), while the simulation-average of the storms obviously cancel each other out, since the simulations have their short-lived storms at different times. We also added a new analysis, summarized in Extended Data Table 3, to highlight that the final year was anomalous, or, to re-use the Reviewer’s phrasing, “not normal”.

Conclusions

Clarify what the SSP levels mean in everyday terms. The discussion assumes the reader is familiar with them. If it’s still acceptable among the climate research community, it might be helpful to label them (e.g. ‘Paris target’, ‘business as usual’ or similar) instead of using the complicated numbering system — otherwise explain what those numbers mean. Relate them to current/expected conditions, if possible.

There are no simple, easily accessible names to refer to the CMIP6 scenarios. But we fully agree with the suggestion to make the conclusion more accessible. We have now added additional explanations to the conclusions to make the text more accessible, by referring the lowest CMIP6 scenario (SSP1-1.9, reaching atmospheric CO₂ concentrations of 393 by 2100, which is below the 2024 value) and similarly adding the atmospheric CO₂ value range in 2100 for the other SSPs mentioned.

The abstract and introduction gave some general motivation for this study, but the conclusions left me asking “why do we care?” The Paris Agreement target of 1.5C is somewhat arbitrary, chosen mainly for galvanizing political support. With regards to impacts of a daily sea ice area level of $1.e6 \text{ km}^2$ and the physical/ecological system motivations mentioned in the introduction, why is this particular number (1.5C) scientifically significant?

We thank the reviewer for pointing this out. To make clear “why we care” about the first ice-free day, we now added a new first paragraph to the final section of the paper. It reads as follows:

“The first time the Arctic will reach ice-free conditions will be an event with a high symbolic significance, as it will visually demonstrate the ability of humans to change one of the defining features of the Arctic Ocean through anthropogenic greenhouse gas emissions - the transition from a white Arctic Ocean to a blue Arctic Ocean (REF). So far, all multi-model predictions have focused on predicting the first ice-free conditions in the monthly (REFS). But the first time we will observe ice-free conditions in the Arctic Ocean will be in the daily satellite data, not in the monthly mean (REF). Thus, to set realistic expectations as to when we could first observe ice-free conditions in the Arctic, we here used daily data from CMIP6 models to provide the first multi-model predictions of the first ice-free day.”

Extended Data Fig 1 - red and pink are nearly indistinguishable

The pink has been changed to light-green to make it clearly visible on screen and in print.

Extended Data Fig 3, 4 - please put the take-away message for the figures in the captions
This has been done for all figures, as suggested by the Reviewer.

Line 75 use “less” instead of “lower”

Corrected

Line 80 use “among” instead of “between”

Corrected

Line 121 isn't the expression “year-round” instead of “year-around”?

This seems to be a UK (year-round) versus US English (year-around) difference, according to the Cambridge dictionary. We'll thus leave it up to the copy editor to decide which one to use. <https://dictionary.cambridge.org/us/dictionary/english/year-round>

Line 166-168 fix sentence grammar. Perhaps change “on” to “in”, “compared to” to “above the”, and put a comma after “level”.

Fixed

Line 170 change “stayed” to “stays” and remove “less than”

Done

Check grammar in the figure captions.

Done

REVIEWERS' COMMENTS

All reviewer comments have been addressed, as detailed below.

Reviewer #1 (Remarks to the Author):

I thank the authors for carefully considering and addressing the comments, and I agree that the analysis is now more rigorous in establishing what factors would contribute to the unlikely but possible scenario of an ice-free day within 6 years. I like that the authors now compare the ice loss in the fast ice loss simulations to observed short-term variability, indicating we could have already produced a RILE under the right circumstances.

I have a few minor comments:

Cooling degree days: The authors might want to check what the correct unit for cooling degree days is. I am not completely sure, but I thought it was cooling degree days, not °C.

The Reviewer is correct. The unit has now been corrected throughout the manuscript.

Note that we changed the acronym from “CDD” to “HDD”. As demonstrated by the Reviewer’s comment, using CDD for “Cumulative Degree Day” was misleading, as CDD usually stands for “Cooling Degree Day”. We switched to the more correct HDD or “Heating Degree Day”.

L. 144-147: I like this part where the authors indicate that a RILE could have already happened in the observed record. I think this sentence is a bit wordy though, and could be shortened to make the message more clear.

As suggested, we have now split the indeed very long sentence into two and shortened the overall statement, to make the message more clear.

Original sentence:

Had we seen a different sequence of events between 2006 and 2012, for example, had we seen the record September sea ice drops leading to the 2007 and 2012 minima within a 4yr period or had we seen a second drop in the September mean SIE as observed between 2006 and 2007 within the 4 year period around 2007, that period would have qualified as a RILE.

The revised sentence (now in two sentences) reads:

A different sequence of observed September SIE minima between 2006 and 2012, however, could have led to a RILE. In particular, a RILE would have been

observed if we had seen the two record September SIE drops leading to the 2007 and 2012 minima within a 4 year period instead of within the 6 year period they occurred in.

Reviewer #2 (Remarks to the Author):

I'm very satisfied with how you have responded to my comments, thank you for the nice work.

There was a a point where I think we had a misunderstanding as I hadn't explained myself well:

For example, if you regress annual-minimum daily SIA against cumulative CO2 emissions (a la Notz and Strove 2016) then you keep an expectation value of daily SIA crossing the million sq km threshold around 2060 with no probability of it reaching this threshold by 2030.

That approach would indeed get us a first ice-free day in the mid 21st century, but that would be the average expected time when we would, due to capturing the forced response (as in Notz and Strove 2016, the focus is on the relationship between the mean SIA and the cumulative emissions). It would not get us the earliest possible dates (the early outer edge of the probability distribution around the average, >2 standard deviations around the mean). As you noted earlier in your comments, it is the rapid nature of the possible transition to ice-free conditions that is important to emphasize, compared to the most likely (=middle of the distribution) timing. Which is why we focus on the quick transition members.

But what I had done was bootstrap the probability of an ice-free day by 2030 based on the observed relationship between annual-minimum SIA and log(CO2), using an AR1 model to simulate natural variability based on observed variability around the trend. In none of the 10,000 members of the bootstrap did I find SIA<1 million sq km by 2030. Hence I questioned whether such RILEs are really feasible. However, I think you discussed very nicely in the text why one may not have observed evidence of such a RILE yet, so I don't see a need for further changes.

Thank you for explaining, we did indeed misunderstand you in the first review cycle.

As the Reviewer wrote that they “don't see a need for further changes”, nothing was changed to the manuscript.

Reviewer #3 (Remarks to the Author):

The authors have addressed my concerns in their revised manuscript and review response.

We thank the Reviewer for taking the time to verify our manuscript again.